# Circulating Tumor Cells as a Promising Tool for Early Detection of Hepatocellular Carcinoma

**DOI:** 10.3390/cells12182260

**Published:** 2023-09-12

**Authors:** Mahsa Salehi, Zohre Miri Lavasani, Hani Keshavarz Alikhani, Bahare Shokouhian, Moustapha Hassan, Mustapha Najimi, Massoud Vosough

**Affiliations:** 1Department of Regenerative Medicine, Cell Science Research Center, Royan Institute for Stem Cell Biology and Technology, Academic Center for Education, Culture and Research, Tehran 1665666311, Iran; s.mahsasalehi@yahoo.com (M.S.); bshokoohian@yahoo.com (B.S.); 2Gastroenterology and Liver Diseases Research Center, Research Institute for Gastroenterology and Liver Diseases, Shahid Beheshti University of Medical Sciences, Tehran 1983969411, Iran; zohreh.mirlavasani@gmail.com; 3Experimental Cancer Medicine, Institution for Laboratory Medicine, Karolinska Institute, 171 77 Stockholm, Sweden; moustapha.hassan@ki.se; 4Laboratory of Pediatric Hepatology and Cell Therapy, Institute of Experimental and Clinical Research (IREC), UCLouvain, B-1200 Brussels, Belgium

**Keywords:** circulating tumor cells, hepatocellular carcinoma, early diagnosis, HCC biomarker

## Abstract

Liver cancer is a significant contributor to the cancer burden, and its incidence rates have recently increased in almost all countries. Hepatocellular carcinoma (HCC) is the most common type of primary liver cancer and is the second leading cause of cancer-related deaths worldwide. Because of the late diagnosis and lack of efficient therapeutic modality for advanced stages of HCC, the death rate continues to increase by ~2–3% per year. Circulating tumor cells (CTCs) are promising tools for early diagnosis, precise prognosis, and follow-up of therapeutic responses. They can be considered to be an innovative biomarker for the early detection of tumors and targeted molecular therapy. In this review, we briefly discuss the novel materials and technologies applied for the practical isolation and detection of CTCs in HCC. Also, the clinical value of CTC detection in HCC is highlighted.

## 1. Introduction

Cancer is a group of disorders in which abnormal cell proliferation and irreversible changes in cellular phenotype result in uncontrolled cell mass growth. Malignancies are the second leading cause of death worldwide [1]. More than 100 types of cancers are known, and they were responsible for an estimated 10.0 million deaths in 2020 [2,3]. Liver cancer is the fourth leading cause of death worldwide and has a high rate of mortality in Asia and Africa. Epidemiological data show that HCC is the fifth most common cancer in men and the seventh in women. Due to the variable prevalence of etiologies, the global incidence of HCC is different, from 72% in Asia to 5% in North America [4]. Currently, therapeutic strategies such as surgery, chemotherapy, radiotherapy, and ignoring inter- and intra-patient heterogeneity are used to treat HCC patients [5,6,7]. Due to the small size and lack of symptoms of the primary tumors, early diagnosis in most types of cancer remains challenging [8]. Therefore, precision oncology using fluid-phase biopsy is indispensable. Liquid biopsy, or fluid-phase biopsy, has promising potential for analyzing the genome basis of cancerous patients, treatment responses, minimal residual disease, and noninvasive therapy resistance [9]. Liquid biopsy is a noninvasive and real-time method used to analyze circulating components such as cell-free DNA (cfDNA) [10], cell-free tumor DNA (ctDNA) [11], extracellular vesicles (EVs) [12], tumor-educated blood platelets (TEPs) [13], and circulating tumor cells (CTCs) [14]. Because of the noninvasive nature of the method, its real-time capability, and molecular heterogeneity, CTCs play a vital role in precision oncology [6]. In spite of their very low concentration in blood and other body fluids, CTCs are promising cell informatics for the diagnosis, prognosis, and follow-up of therapeutic responses [15]. Therefore, CTCs can be considered as biomarkers for the early detection of tumors, targeted molecular therapies for cancer patients, and for determining CTC phenotypes in preclinical models [15,16,17]. In the clinic, it was shown that CTC-based micro-devices could be an ideal modality for point-of-care testing. However, due to their heterogeneity, reliable detection of CTCs in body fluids is still a major limitation. Indeed, CTCs derived from different tissues have various characteristics, such as different sizes, markers, and immune-phenotyping profiles, which make their detection more challenging. Furthermore, several other limiting factors such as damage and fragmentation, both in vivo or in vitro during the isolation process, hamper their clinical application [18]. In this review, we briefly discuss the novel materials and technologies for their isolation and early detection. Also, the clinical value of their detection in cancers is highlighted.

## 2. Biology of Circulating Tumor Cells

CTCs were described for the first time by Ashworth in 1869 as cells in the blood of metastatic cancer patients with similar properties to the primary tumors [19]. CTCs represent a small fraction of the cells in the blood and are defined as cancer cells that have departed from a solid tumor lesion and entered the bloodstream [20]. CTCs are found in the bloodstream of patients as isolated CTCs (iCTCs) or as circulating tumor microemboli (CTMs) [21]. Some experiments have supported that tumor cells can spread even during the early stages of evolution [22,23]. Despite their origin, CTCs are distinct from primary tumor cells [24]. CTCs gain the epithelial-to-mesenchymal transition (EMT) potential that helps them dissociate from the primary tumor and facilitates their entry into the bloodstream. CTCs can disseminate from the cell clusters and exhibit stemness features that increase their metastatic potential [24,25]. It is worth mentioning that most CTCs are eliminated in the circulation, and only a few of them survive and reach the other organs [25]. The molecular characteristics of CTCs in the early stages of tumor evolution could be a promising tool for early diagnosis and the prevention of metastasis. A panel of CTCs’ molecular markers can be used to track these cells in the circulation. The vast majority of such markers are those related to epithelial markers, such as the epithelial cell adhesion molecule (EpCAM) [26]. EMT-related molecules can also be used. During the EMT process, the expression of epithelial markers such as E-cadherin, ZO-1, claudins, and occludins decreases, while the expression of mesenchymal markers, including vimentin, N-cadherin, fibroblast-specific protein 1, and fibronectin, increases [27]. EMT-related transcription factors such as SNAIL and the zinc finger E-box-binding homeobox (ZEB) families can be used as a marker, but because of their cytoplasmic or nuclear origin, they are not currently available for CTC detection [27]. EMT-related proteins such as E-cadherin, vimentin, and TWIST are accessible markers, and thus were analyzed using flow cytometry sorting, immunostaining, and fluorescence in situ hybridization (FISH) staining technologies to track the CTCs. Nevertheless, single-cell CTC sequencing technologies can be used at the RNA level to cover all the EMT-related markers discussed [28]. Based on different cancer types, other CTC biomarkers, including estrogen receptor [29], folate receptor [30], human epidermal growth factor receptor-2 (HER2) [31], prostate-specific membrane antigen [32], and survivin [33], have been used in the clinic [25]. Due to the clonal selection of CTCs or clonal acquisition, there is relative discordance in the expression of some markers between the primary tumor and the CTCs [25].

### CTC Entry into the Circulation and Metastasis

To establish the metastasis process, cancer cells must move from the primary tumor into the bloodstream, access the target tissue, colonize, and ultimately grow in the secondary tissue. Indeed, CTCs are an intermediate stage of metastasis, and can actively or passively access the bloodstream. CTCs can circulate in the bloodstream as single cells or clusters. The cluster cells have an increased metastatic potential, while the single cells have a longer half-life [34]. Due to physical and oxidative stress, a lack of growth factors and cytokines, as well as anoikis, most CTCs in the bloodstream cannot survive, but a few remain alive, actively extravasate into the target tissue, and begin to divide and colonize [22]. EMT is the main hypothesis for the intravasation process of tumor cells [35]. In the primary tumor, EMT facilitates intravasation into the bloodstream and increases the migration potential of cancer cells. Moreover, when the cancer cells are covered with platelets in the bloodstream, the EMT phenomenon may occur [36]. The reverse process, mesenchymal to epithelial transition (MET), takes place when the CTC cells extravasate and continue to proliferate in the secondary organ [35]. Current knowledge about the mechanisms of CTC generation and their intravasation, circulation, extravasation, proliferation, and colonization are summarized in Figure 1.

## 3. Novel Strategies for CTCs’ Isolation in HCC

CTCs have undoubtedly a great clinical significance and can reflect valuable information in the diagnosis, prognosis, and response to treatment of cancer patients. However, their extraordinary rarity in the bloodstream has created an important challenge in their study and the evaluation of their yield [37]. In HCC patients, only a range of 0–86 CTCs were detected in 5 mL of blood [38]. Therefore, technologies with high specificity and sensitivity are required to capture CTCs for downstream analysis. In recent years, several technologies have been employed; these are generally classified into two categories, label-dependent and label-independent methods, which are discussed below.

### 3.1. Label-Dependent Strategies

Label-dependent methods are among the most commonly used techniques to isolate CTCs, and rely on the interaction of affinitive agents (e.g., antibodies or aptamers) tethered on device surfaces or magnetic beads with cell membrane markers [39]. Positive and negative enrichments are two main subcategories of label-dependent techniques. Positive enrichment methods are based on the use of antibodies against tumor-specific antigens that are present on the membrane of the CTCs. Therefore, these techniques can directly isolate CTCs. Inversely, the negative enrichment methods use antibodies that bind to antigens on the surface of blood cells (e.g., CD45), leading to the removal of nonspecific cells and the enrichment of the CTCs indirectly [40].

To date, various specific tumor markers, including HER2, PSA, EGFR (epidermal growth factor receptor), EpCAM, and MUC1 (mucin-1) have been used to isolate CTCs; among these, EpCAM has been extensively used [41,42]. This marker is often expressed in cells with epithelial origins and is not present in blood cells. Therefore, it can be used to isolate CTCs originating from epithelial tumors (e.g., HCC) [43]. Racila et al. reported that cancer cells could be detected in the circulation by using immunomagnetic and flow cytometry techniques [44]. They detected CTCs based on their expression of EpCAM and non-expression of CD45. These markers would become the basis for the initial definition of CTCs and were used to develop some CTC detection methods like the CellSearch^®^ System (Figure 2) [45]. CellSearch^®^ is an example of an EpCAM-affinity-based platform for CTCs’ enrichment. This system is the only FDA-approved method for capturing CTCs that utilizes ferrofluid nanoparticles conjugated with anti-EpCAM antibodies to capture CTCs in 7.5 mL blood samples. Isolated CTCs are then fixed and stained with DAPI and a cocktail of fluorescence-tagged antibodies against epithelial cytokeratins (CKs) and CD45 (leukocyte-specific marker) markers. DAPI^+^/EpCAM^+^/CK^+^/CD45^−^ cells are considered to be CTCs, while DAPI^+^/CD45^+^ cells are considered to be leukocytes [45]. In 2015, Kelley et al. used the CellSearch System to isolate EpCAM-positive CTCs in metastatic HCC samples [46]. In another study conducted by Sun et al., the prognostic value of CTCs isolated with the CellSearch^®^ System in HCC patients undergoing curative resection was investigated, and EpCAM-positive CTCs were detected in 66.67% of patients prior to resection [47]. The conjugation of EpCAM-affinitive agents to magnetic beads, and then the collection of the captured CTCs through a magnetic field, is another popular method that has been used to enrich CTCs from the blood of patients [39]. In 2015, Pilapong et al. developed magnetic nanoparticles conjugated to anti-EpCAM DNA-based aptamers to isolate CTCs related to HCC [48]. However, the lower sensitivity and recovery rate of CTCs significantly limited its wide clinical application [24].

Although many studies highlighted EpCAM as a suitable marker to be used for isolating CTCs in HCC, only 35% of all HCC cases were positive for this marker, which can significantly reduce the sensitivity of this method [49]. Moreover, during the EMT process in the metastatic cascade, the expression of EpCAM is considerably inhibited, leading to the escape of CTCs with highly metastatic properties from the EpCAM-based isolation systems [26,50]. Due to the limitations of the epithelial marker EpCAM, the application of positive selection strategies that target mesenchymal (e.g., vimentin or N-cadherin), stem cell (e.g., CD133), or tumor-specific markers can be beneficial [40]. The asialoglycoprotein receptor (ASGPR) is a transmembrane protein commonly found on the surface of hepatocytes and HCC cells. Xu et al. developed a system that isolates CTCs of the HCC that is based on the interaction of ASGPR with its ligand [51]. In this system, the cells were first bound to the biotinylated ligand of ASGPR and then were magnetically separated through anti-biotin antibody-conjugated magnetic beads. In another study, researchers developed anti-ASGPR antibody-coated magnetic beads that could detect CTCs with high sensitivity and specificity in HCC [52]. Compared with the previous system based on receptor–ligand interaction, this system had a higher capturing efficiency. Although this approach is a better strategy to overcome the low sensitivity of EpCAM-based systems, capturing CTCs with a single marker is almost ineffective due to the heterogeneous nature of HCC; therefore, a combination of markers is needed. In 2018, a novel subtraction enrichment immunostaining-fluorescence in situ hybridization (SE-iFISH) strategy was developed to detect the HCC-CTCs. This technique was based on the comprehensive detection of in situ phenotypic and karyotypic characterization of hepatocellular CTCs (CD45−/CD31−) in patients subjected to surgical resection [53,54]. In addition, a microgravity array (MCA) system was also used to detect CTCs and their mRNA expression in HCC patients [55].

Apart from magnetic nanoparticles, other functionalized nanostructured materials have also been used for the isolation and detection of CTCs. In 2016, Wang et al. designed a chip (CTC^−BioT^Chip (to isolate CTCs from HCC [53]. In this chip, a hydroxyapatite/chitosan nanofilm that has a good cell-preferred nanoscale topography and is coated with cell-surface carbohydrate sialyl Lewis X was used to improve the capturing of CTCs. In another study in 2019, Wu et al. created a reduced graphene oxide film for CTC detection that was modified with an anti-EpCAM antibody and galactose-rhodamine-polyacrylamide nanoparticles. The fluorescence was quenched with a reduced graphene oxide film [54]. In this system, CTCs were first captured with anti-EpCAM antibodies, and then galactose-rhodamine-polyacrylamide nanoparticles were endocytosed into CTCs by the ASGPRs present on their surface, resulting in high fluorescence recovery. This platform could detect as few as five CTCs in 1 mL of the spiked blood sample. The volume of blood needed for CTC analysis using CTC^−BioT^Chip depends on factors such as the device design, channel dimensions, and the specific application. Typically, the recommended sample volume ranges from a few microliters to milliliters, depending on the specific experimental requirements.

Microfluidic systems have also been successfully used to isolate CTCs from HCC based on a positive selection strategy [55]. Zhang et al. developed a microfluidic chip that provided the isolation of viable CTCs. The channels of this system had small dimensions that facilitated the local topographic interactions. Furthermore, because the channels were coated with an ASGPR ligand, asialofetuin, efficient capturing (>85%) of CTCs related to HCC was achieved [56]. In 2018, Court et al. used a microfluidic chip, NanoVelcro, coated with a cocktail of antibodies targeting surface markers including ASGPR, glypican-3, and EpCAM (Figure 2) [57]. This microfluidic system provided enhanced topographic interactions while coated antibodies enabled the efficient capturing of CTCs. This platform could detect CTCs in 97% of patients. The required sample volume for CTC analysis in microfluidic systems ranges from microliters to milliliters. The exact volume depends on various factors such as the device design, channel dimensions, and sensitivity. Smaller microfluidic devices may require lower sample volumes, while larger devices or high-throughput systems may handle larger volumes [58].

Although the positive selection strategy is an effective approach for the isolation of CTCs, due to the heterogeneity of the tumor, even a combination of antibodies targeting different cell surface antigens may not be a suitable approach. Negative selection strategies can solve this issue by depleting background blood cells using an anti-CD45 antibody [40]. Therefore, this method is suitable for downstream analyses such as genetic assays, CTC culture, and xenografts [59,60]. Currently, this approach has been successfully applied in several studies to isolate HCC-CTCs. Liu et al. utilized magnetic beads coated with anti-CD45 antibodies to extract leukocytes and enrich CTCs [61]. Compared with the ASGPR positive-selection strategy, this method could achieve a higher recovery of spiked HCC cells. Despite the higher sensitivity of the negative selection strategies than the positive enrichment techniques, these methods still suffer from much lower purity [40].

### 3.2. Label-Independent Methods

Unlike label-dependent techniques, which rely on surface markers to capture CTCs, label-independent methods use the differences in the physical properties of CTCs and blood cells (e.g., size, density, etc.), thus preventing the escape of CTCs that do not express a specific marker [37,42]. Furthermore, because isolated CTCs are not bound to any antibody, they are easier to process for downstream applications [42].

#### 3.2.1. Size and Deformability

One of the most popular methods for isolating CTCs is the size-based technique. These methods typically benefit from the size differences between CTCs and normal hematological cells because most CTCs are larger than normal hematological cells [62]. So far, different types of filters with different materials have been developed to capture CTCs. Isolation by size of tumor cells (ISET^®^) is a filtration-based technology developed in 2000 by Paterlini-Bréchot and her colleagues to isolate CTCs (Figure 2) [63]. This system has been successfully used for the enrichment of CTCs from blood samples in liver cancer patients [64]. In 2014, Morris et al. compared the ability of CellSearch^®^ and ISET^®^ to detect CTCs from HCC [65]. Using the ISET^®^ system, CTCs were identified in 100% of HCC patients, while CellSearch^®^ identified CTCs in only 28% of cases. The ISET^®^ system generally requires a volume of blood ranging from 5 to 10 mL for CTC analysis. This volume is processed using specialized ISET^®^ filtration devices and following associated protocols [66]. Although filtration allows the rapid and convenient isolation of CTCs, it still faces challenges. CTCs are highly heterogeneous; whereas some CTCs are bigger than blood cells, others are the same size as or even smaller than circulating leukocytes. Therefore, this technique is able to isolate only CTCs larger than blood cells and is often unable to enrich smaller CTCs [42]. Another challenge is the regular application of tumor cell lines in the initial validation process. Because CTCs from clinical samples of cancer patients are significantly smaller than cancer cell lines, results are irrelevant when cells from tumor cell lines are used for validation [67]. Furthermore, filtration systems face clogging, and their intense tension can affect the viability of the CTCs [40].

Deformability is another physical characteristic that has been used for CTC isolation. Bagnall et al. compared the deformability of CTCs and normal hematological cells [68]. They showed that CTCs and WBCs have different deformability. The advantages of this approach are simplicity and low cost. Parsortix™ is a microfluidic-based system that traps CTCs based on their size and deformability. In this technology, whole blood passes through the filtration cassettes, and CTCs are trapped in the cassettes based on their different size and deformability compared with other blood components [69]. This system can isolate viable CTCs usable for downstream analysis (Figure 2). The Parsortix™ system typically requires a volume of blood ranging from 5 to 20 mL for CTC analysis [69].

#### 3.2.2. Density

Density gradient centrifugation is the early technique reported to isolate CTCs. Indeed, a sample with different cell populations is subjected to centrifugation, the different types of cells pass through the density gradient, and each is suspended at the point where its density equals the surrounding medium. Therefore, following blood density gradient centrifugation, CTCs can be separated from the denser cells [70]. In 1950, Fawcett et al. used albumin density centrifugation to isolate tumor cells from other cells in peritoneal fluid [71]. However, using albumin as a flotation medium was expensive and difficult to prepare. In 1959, Seal et al. used silicon blending oil as a biologically inert and inexpensive floating medium to isolate CTCs from blood samples [72]. Various density gradient media like Percoll and Ficoll have also been developed, each of which has its own strengths and weaknesses. OncoQuick^®^ is one of the novel density gradient centrifugation-based technologies designed for CTC isolation (Figure 2) [73]. In this system, a porous membrane is placed in the 50 mL polypropylene centrifugation tube, which prevents mixing of the sample with the separation medium located in the lower compartment. Following centrifugation, cells are separated based on their buoyant density. Accordingly, denser cells (e.g., red blood cells (RBCs) and granulocytes) pass through the membrane and enter the lower compartment. At the same time, CTCs remain in the interphase layer formed between the plasma and the separation medium in the upper compartment. RosetteSep™ CTC enrichment cocktail, STEMCELL Technologies Inc., Vancouver, BC, Canada, is another density gradient centrifugation-based platform that enriches CTCs by combining this method with antibodies to eliminate unwanted cells [37]. OncoQuick^®^ typically requires a blood sample volume ranging from 1 to 10 mL for CTC analysis [37]. This strategy has been successfully applied to capture CTCs from HCC (Figure 2) [74].

Although density gradient centrifugation is one of the most widely used strategies to isolate CTCs, due to its low sensitivity and the risk of contamination with other cells, this method is usually used in combination with other applications to increase the purity of the recovered CTCs [39]. For example, Guo et al. used Ficoll density gradient centrifugation followed by sequential incubation of the peripheral mononuclear blood cells with anti-CD45 and anti-Ber-Ep4 antibody-coated magnetic beads for further enrichment [75]. In another study in 2019, Hamaoka et al. used density gradient centrifugation along with an immunogenetic positive enrichment method to capture glypican-3 (GPC3)-positive CTCs in the HCC blood samples [76].

## 4. Molecular Analysis of CTCs

Due to the vast degree of heterogeneity, variability in isolation technologies, potential biases during downstream molecular processing, and lack of reproducibility from one study to another, CTC clinical utilities remain relatively limited. Abundant molecular data derived from genomic, transcriptomic, proteomic, and metabolomic levels could help in developing specific CTC-based biomarker panels and, therefore, in disease monitoring [77]. Nevertheless, in bulk tumors, only average profiles of different sub-clones have been reflected. The heterogeneity landscape of HCC cells at single-cell resolution remains also largely unknown. Deep sequencing of somatic mutations to enumerate copy number variation at the level of single cancer cells has led to an increased recognition of intra-tumor heterogeneity (ITH) during cancer progression. The results of a study using the CanPatrol^TM^ technique provided evidence for the CTC-WBC cluster as a potential predictor of disease-free survival (DFS), overall survival (OS), and poor prognosis of HCC [78]. Although various molecular markers have been used to detect CTCs in HCC, with the aim to improve prognosis and treatment selection, accurate biomarker identification is a critical “unmet need” [40]. In this regard, a novel multi-marker CTC enrichment assay with high efficiency and accuracy has been developed [79]. To enhance the capture efficiency, a synergistic chip with a deterministic lateral displacement (DLD)-patterned microfluidic design is employed, which effectively combines the complementary effects of anti-ASGPR and anti-EpCAM antibodies. This strategic alignment of the antibodies on the chip aims to maximize the efficiency of capturing the target cells. The CTC-capture optimizing was 100% (45 out of 45) in HCC patients, with 97.8% and 100% sensitivity and specificity, respectively [80]. Over the last decade, advances in molecular methods have generated a range of successful strategies for the analysis of single CTCs [81]. The application of cell-free DNAs (cfDNAs) as molecular targets can be used for the detection of HCC-CTCs. cfDNA-based technologies have several advantages, such as real-time monitoring of the genetic landscape of the tumor, high sensitivity and specificity, and being a noninvasive biopsy procedure. However, the application of this technology has faced some challenges, including ensuring the purity of cfDNA samples, the genetic heterogeneity of tumors, and the development of standardized protocols for the analysis of cfDNA data [82]. In Table 1, the methods for single-cell analysis at the molecular level and their advantages and disadvantages are briefly described.

### 4.1. Genomic Level

Cancer cells in the tumor may be from the euploid, pseudoeuploid, or aneuploid subpopulations. Using the next-generation sequencing (NGS) and single-cell sequencing (SCS) technologies, it is now possible to decipher the complete genomes of CTCs [83]. The whole-genome amplification (WGA) method yields accurate genomic analysis of CTCs (Table 2). WGS, whole-exome sequencing, or targeted sequencing, which are examples of improved WGA methods, could further decrease sequencing requirements and enable more cost-effective interrogation of all genomic variations in single cells, including single nucleotide variants (SNVs) and structural variants that reside in noncoding regions [77]. Novel WGA methods, primary template-directed amplification (PTA), and multiplexed end-tagging amplification of complementary strands (META-CS) have been developed to decrease false positives and increase the accuracy of SNV indications. In addition, droplet digital PCR (ddPCR) has been proposed as a novel method to achieve amplification throughout the genome. By partitioning the DNA sample into numerous individual droplets and performing PCR amplification in each droplet separately, ddPCR enables more precise quantification of target DNA sequences. This approach reduces amplification bias and provides more uniform representation of the entire genome, resulting in more accurate and reliable results. Recently, a novel filtering-based microfluidic technology at the single-cell level in a chip was developed to minimize cell loss and potential cellular cross-contamination [84].

In 2011, the American Anderson Cancer Research Center and Cold Spring Harbor Laboratory developed single-cell sequencing analysis technology. To date, some technologies including NGS, Sanger sequencing, array comparative genomic hybridization (aCGH) platforms, single-nucleotide polymorphism (SNP), and conventional PCR technologies were developed to analyze somatic SNVs, structural variations (SVs), copy number variations (CNVs), and chromosomal breakpoints and rearrangements for the whole exome/genome or selected cancer-associated genes [85]. Single-cell DNA sequencing (scDNA-seq) was performed on cells isolated from 10 patients with HCC, and ploidy-resolved scDNA sequencing was performed on the cancer cells of one additional patient. The results of the scDNA-seq analysis revealed that the copy number alterations in HCC are followed by dual-phase copy number evolution. In fact, patients with prolonged gradual phases have higher intra-tumor heterogeneity. This study’s results also showed that the *CAD* gene involved in pyrimidine synthesis has an important role in tumorigenesis. The results of ploidy-resolved scDNA sequencing demonstrated that the doubling of diploid tumor cells is a common way of generating polyploid tumor cells in HCC [86]. In an experiment, a 10-gene CTC signature was used to evaluate the therapeutic efficacy on HCC patients. The results showed that this method can be useful for the early detection of HCC in a high-risk population [77]. Branched-chain amino acid transaminase 1 (*BCAT1*), as the identified biomarker gene in the HCC, was shown to be significantly upregulated or knocked down in HepG2, Hep3B, and Huh-7 cells, leading to a reduction in cell proliferation, migration, and invasion or apoptosis. It was shown that the increase in EpCAM and E-cadherin expression and the reduction in vimentin and TWIST expression suggest that *BCAT1* may trigger the EMT process. *BCAT1* overexpression may induce CTC release by triggering EMT and may be an important biomarker of HCC metastasis (ST) [84]. The top mutated genes in stage I of HCC are *TP53*, *CTNNB1*, *TTN*, *MUC16*, and *ALB*, and their co-mutations or mutually exclusive mutations were identified in HCC. Currently, 29 genes are identified with significant roles in prognosis, including highly mutated *LRP1B*, *ARID1A*, and *PTPRQ* genes. It was shown that for the patients with wild-type genes, overall survival rates are significantly better than those with mutant ones. Patients in the top 10% of the tumor mutational burden (TMB) exhibited significantly worse prognoses than the other 90% (ST) [55].

Recently, Yi et al. introduced a specific technique for the enrichment of HCC-CTCs using glypican-3 immuno-liposomes (GPC3-IML). The results of this study showed that GPC-3 could be used as a more reliable CTC isolation biomarker than EpCAM and vimentin. Positive correlation was observed between the count of CTCs (≥5 PV-CTC per 7.5 mL blood) and BCLC stage (*p* = 0.055). The result of the CTC-NGS was consistent with that of tissue-NGS in 60% of the cases, revealing that *KMT2C* is a common, frequently mutated gene [87]. Currently, some challenges in the genomic analysis of CTCs remain to be solved, including high genome coverage, low allele dropout, and low amplification errors [88]. However, despite all of these limitations, single-CTC genomic analysis could be a powerful noninvasive diagnostic tool to investigate the changes in the gene expression profiles of cancer patients with localized, metastatic, and recurrent diseases.

### 4.2. Transcriptomic Level

Recently, scientists developed some highly sensitive and specific molecular CTC assays using microfluidic enrichment of CTCs coupled with digital-droplet PCR (ddPCR)-based profiling technologies [89]. Over the past four years, some liver scRNA-seq studies showed that a combination strategy of scRNA-seq and smRNA-FISH could be used to obtain spatial information. These experiments showed that following bioinformatic protocols and specific sequencing strategies can integrate each cell’s RNA data with spatial information [90]. In 2014, a refined platform based on mRNA isolation and cDNA synthesis methods in comparison with CellSearch^®^ was designed. This prospective study, which included 299 patients with HCC, was completed and indicated the qRT-PCR-based CTC detection method was significantly preferable in regard to sensitivity, specificity, reproducibility, and the small sample size required. This system was proposed for adjuvant diagnosis, assessment of therapeutic response, and prompt decision-making to adopt the most effective antitumor strategies [74].

During the last decade, a CanPatrol^TM^ CTC-enrichment technique based on RNA in situ hybridization (RNA-ISH) has been reported. It uses both epithelial and mesenchymal markers such as EpCAM, CK8/18/19, E-cadherin, vimentin, and TWIST for the characterization and classification of CTCs into all three CTC subpopulations in different types of cancer [91]. EpCAM, CK8/18/19, TWIST, and vimentin are common EMT markers and were evaluated using FISH through the CanPatrol^TM^ enrichment platform in many HCC-related studies [92]. Using CanPatrol^TM^ and an in situ hybridization technique, Qi et al. demonstrated that the suppression of BCAT1 reduced HCC cell proliferation, migration, and invasion and promoted apoptosis, probably by inhibiting EMT. Additionally, 67 differentially expressed cancer-related genes (DEGs) involved in cancer-related biological pathways were identified [93].

High expression of *CAD*, a gene involved in pyrimidine synthesis, is correlated with rapid tumorigenesis and reduced survival in HCC patients. The results integrating bulk RNA-seq of 17 HCC patients, published datasets of 1196 liver tumors, and immuno-histochemical staining of 202 HCC tumors confirmed these results [86]. Furthermore, it was shown that in stage I of HCC, some parameters, including cell skeleton proteins, ion channels, cell cycle, etc., are dramatically changed. Some independent risk factors related to HCC such as *MMRN1*, *OXT*, and *COX6A2* transcription; sex; race; etc. are used to predict the prognosis of the disease [94], while mutational and transcriptional alterations and clinicopathological factors could predict the prognosis of stage I HCC. Analyzing the whole-exome somatic mutation data, whole mRNA transcription data, along with demographic and clinical information from the TCGA database, could also be helpful [94]. Yao et al. used single-cell RNA sequencing technology and showed that some phosphorylation-related genes such as *POLR2G*, *PPP2R1A*, *POLR2L*, *PRC1*, *ITBG1BP1*, *MARCKSL1*, *EZH2*, *DTYMK*, and *AURKA* are highly expressed in HCC [95]. In these experiments, ingenuity pathway analysis revealed two hub genes, *AURKA* and *EZH2*, with high expression in HCC malignancy, which suggests that an AURKA inhibitor (alisertib) and an EZH2 inhibitor (gambogenic) could be used for the inhibition of HCC cell proliferation, migration, and invasion [95]. Table 3 summarizes all these technological approaches developed for the detection and the isolation of HCC-CTCs.

### 4.3. Proteomic Level

The ability to perform multiplexed protein analysis targeting CTCs offers a unique and valuable opportunity to gain additional insights into CTC biology. This approach enables researchers to perform simultaneous analysis of multiple proteins and achieve a more comprehensive understanding of the characteristics and behavior of individual CTCs. By providing specific and precise information, this method contributes to expanding our knowledge of CTC biology and assessing its clinical significance. Currently, the EPithelial ImmunoSPOT (EPISPOT) assay, also called EPISPOT in a DROP (EPIDROP), has been used to analyze the proteome and secretome data of viable CTCs simultaneously [85]. The content and expression level of the TWIST and vimentin proteins in CTCs could be used as biomarkers for evaluating metastasis in HCC. ASGPR, which can be stained using immunofluorescence techniques, is another protein marker to be used for the detection of CTCs. Li et al. used the triple-immunofluorescence staining method to detect the expression of TWIST and vimentin in the CTCs obtained from 39 (84.8%) and 37 (80.4%) of the 46 analyzed patients, respectively. Also, co-expression of these two proteins could be detected in 32 (69.6%) of the 46 patients [96]. Using the NanoVelcro CTC assay, an antibody cocktail targeting the cell-surface markers such as ASGPR, glypican-3, and epithelial cell adhesion molecule has been optimized to capture the CTCs of HCC [57]. C-X-C chemokine receptor type 4 (CXCR4) and matrix metallopeptidase 26 (MMP26^+^)-positive CTCs were also considered as markers for detecting CTCs in liver cancer [97].

### 4.4. Epigenomic Level

Epigenetic changes are the leading cause of tumor cell transformation and progression of cancer. Histone modifications, DNA methylations, and miRNA-mediated processes are major epigenetic changes that are critically associated with various mechanisms of proliferation and metastasis in several types of cancer [81]. DNA methylation is the major epigenetic change in cancer cells and could be used as a biomarker for the detection of CTCs. It was confirmed that the methylation patterns during tumorigenesis are not randomly organized [86]. DNA methylation remodeling as an important epigenetic change has been widely observed in several genes involved in EMT, tumor cell dissemination, and the acquisition of stem cell properties that are crucial for CTCs [4,95]. Methylation and epigenetic changes in genes encoding E-cadherin (*CDH1*), TWIST, vimentin (*VIM*), N-cadherin (*CDH2*), and the miR-200 family of miRNAs have also been confirmed by further analyses [96]. Hypermethylated genes found in HCC, such as *CDKN2A*, *RASSF1*, *APC*, and *SMAD6*, are among the good markers for the detection of CTCs in these kinds of cancers [83]. Eleven methyltransferases and demethylases, including enhancer of zeste homolog 2 (EZH2), euchromatic histone-lysine N-methyltransferase 2 (EHMT2), SET domain bifurcate 1 (SETDB1), and SET domain 2 (SETD2), were found that play a role in the clinical stages of HCC, which confirms the fundamental role of histone methylation regulation in HCC progression [88].

**Table 3 cells-12-02260-t003:** Different platforms for the detection and isolation of CTCs and their markers.

Platform	Analyses	Marker(s)/Parameter	References
CellSearch^®^	Anti-EpCAM antibody Immunohistochemistry (IHC)-based approach	EpCAM	[44,98]
NanoVelcro	Microfluidic chip coated with a cocktail of antibodies	Surface markers including ASGPR, glypican-3, and EpCAM	[99]
CTC^−BioT^Chip	Hydroxyapatite/chitosan nanofilm	EpCAM	[53]
Refined CTC^−BioT^Chip	Anti-EpCAM antibody and galactose-rhodamine-polyacrylamide nanoparticles were endocytosed into the CTCs through ASGPRs present on the surface of the CTCs.	EpCAM and ASGPR	[53]
ISET^®^	Filtration-based technology	Cytokeratin (CK)	[100]
Parsortix	Microfluidic-based system	Size and deformability	[69]
RosetteSep	Density gradient centrifugation-based platform	Cocktail antibody	[101]
OncoQuick	Density gradient centrifugation-based technologies	Buoyant density	[73]
CanPatrol^TM^	Microfiltration and various EMT markers	EpCAM, CK8/9/19, vimentin, and TWIST	[78]
EP@MNPs	Novel peptide-based magnetic nanoparticle	EpCAM recognition peptide followed by CD profiling to distinguish epithelial and mesenchymal subgroups	[102]
NP@MNPs	Novel peptide-based magnetic nanoparticle	N-cadherin recognition peptide followed CDRNA profiling to distinguish epithelial and mesenchymal subgroups	[103]
CytoSorter^®^ and CytoSorter™	CTC PD-L1 Kit	PD-L1 antibody	[104]
Optimized CanPatrol CTC-enrichment	Combining nanotechnology filters and mRNA ISH array	EpCAM, CK8, CD18, CD45, Vimentin, TWIST, CK19, and NANOG	[105]
EPIDROP	Single-cell proteomic and secretomic analyses of viable CTCs	EpCAM and other IHC markers	[85]
RT-LAMP	Reverse transcription loop-mediated isothermal amplification	EpCAM, CK19, CD133, and CD90	[106]
RareCyte	High-definition single-cell analysis (HD-SCA)	EpCAM, CK, and other IHC markers	[107]
DEPArray™	Sub-sequential high-quality genomic profiling	A combination of dielectrophoresis (DEP) and image-based selection methods and some IHC markers	[108]
NanoVelcro	Triple-immunofluorescence staining method	ASGPR, glypican-3, and epithelial cell adhesion molecule	[57]

## 5. Clinical Value of CTC Detection in HCC

Currently, CTCs can be used as alternative biomarkers for the early detection of HCC. Some studies have investigated the potential applications of CTCs in early detection of HCC, but no clinical guidelines are currently included in routine clinical use [25]. CTCs play a crucial role in the initiation of metastasis and therefore were suggested as biomarkers for the early detection of HCC; EMT-related markers are useful for the early diagnosis and staging of those cancer cells. In a study, the results showed that 30.5% (18/59) of HCC patients have EpCAM^+^ CTCs, while this marker was found in only 5.3% (1/19) of individuals in the control group of patients with cirrhosis or benign hepatic tumor (Table 4). EpCAM^+^ CTCs were also used for staging the HCC, as significant differences in CTC detection rates were observed in different Barcelona Clinic Liver Cancer (BCLC) stages [109]. Qi et al. showed that 101 out of 112 patients diagnosed with HCC had positive detection of CTCs. Remarkably, the presence of CTCs was observed not only in advanced stages but also in the early stages of the disease [93]. The number of peripheral blood mesenchymal CTCs was high in late-stage HCC patients, for example in the B–C stages of BCLC; therefore, mesenchymal CTCs are the cut-off value for the diagnosis of BCLC stage in HCC patients [110]. Indeed, Li et al. showed that the EMT biomarkers such as TWIST and vimentin could serve as promising targets for evaluating metastasis and prognosis in HCC patients [96].

HCC-specific markers could be efficiently used for the early and specific detection of HCC-CTCs in the clinic. ASGPR is exclusively expressed in the human hepatoma cell line, normal hepatocytes, and HCC cells. Anti-ASGPR antibodies could efficiently detect the circulating HCC cells. Also, the antibody cocktail against carbamoyl phosphate synthetase 1 (CPS1) and pan-cytokeratin (P-CK) has been demonstrated to detect CTCs in HCC [52].

Currently, there is no approved data supporting the usefulness of CTCs as an early-stage HCC diagnostic tool. Nevertheless, many investigations have suggested that CTCs can be helpful in predicting the therapeutic outcome and monitoring the disease progression, particularly after resection. Furthermore, the diagnostic value of different phenotypes of CTCs in HCC has been evaluated. CanPatrol™ technology using the microfiltration system was developed for the isolation and characterization of the different CTC types, including epithelial (EpCAM and CK8/9/19), mesenchymal (vimentin and TWIST), mixed, and total CTCs [40]. In a study, the CanPatrol™ CTC-enrichment technology was employed in a cohort study of 112 HCC patients. Analysis of the collected data uncovered that CTCs were detected in more than 16 patients. Furthermore, they showed that the proportion of mesenchymal CTCs exceeded 2% in them (Table 4) [93].

Liquid biopsy to detect prognostic and predictive biomarkers can be a potential avenue for improving early diagnosis and more efficient treatment for HCC. Specifically, various genetic alterations and molecular changes have been investigated in CTCs for their utility in diagnosing and managing HCC. These include copy number variations; gene integrity; mutations in *RAS*, *TERT*, *CTNNB1*, and *TP53* genes; as well as DNA methylation changes in *DBX2*, *THY1*, and *TGR5*. Furthermore, the signaling pathways associated with certain biological functions, such as the MAPK/RAS pathway, p53 signaling pathway, and Wnt-β catenin pathway, have been explored and emphasized in the context of HCC diagnosis and management. Employing these approaches can provide valuable insights into HCC progression, allowing for early detection and the optimization of treatment strategies. By monitoring these molecular alterations in CTCs, liquid biopsy holds great promise as a noninvasive tool for precise prognosis and guiding personalized treatment decisions for HCC patients.

## 6. Conclusions

CTCs and related technologies are promising tools for both diagnostic and prognostic applications of the early stages of HCC. Currently, many advanced technologies are used for the detection of CTCs in the clinic, but still, there are significant challenges that need to be addressed. These technologies, along with added values from bioinformatics and annotated databases, could be optimized to efficiently track and detect CTCs in the blood and to perform risk assessment in this regard as a novel diagnostic approach. Moreover, the combination of detection methods using CTCs and cfDNAs can be a promising approach for early diagnosis. It was shown that the sensitivity and specificity of current CTC biomarker panels require substantial improvements. Indeed, novel ultra-high-throughput quantification strategies are needed to analyze simultaneous profiles of multi-marker panels and to provide comprehensive coverage of the highly heterogeneous cancer cell subpopulations. These shortcomings necessitate the adoption of a comprehensive interdisciplinary approach and precise devices for analyzing large datasets. Hopefully, machine learning can facilitate CTC-related assessments and validate perspective results.

## Figures and Tables

**Figure 1 cells-12-02260-f001:**
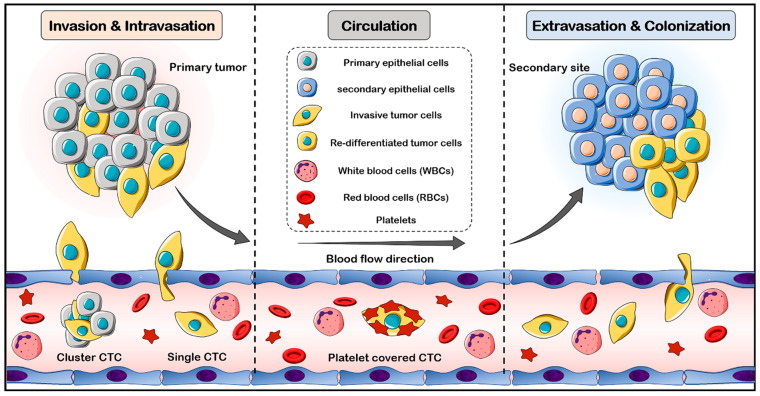
Schematic representation of CTC generation, intravasation, circulation, extravasation, proliferation, and colonization. In the invasion process, cancer cells actively break down the basement membrane, migrate through the extracellular matrix, and enter the circulation. CTCs detach from primary tumors in the form of both single cells and cell clusters. Once the CTCs enter blood circulation, they are covered with platelets, and this coating induces the EMT phenotype in the CTCs. In secondary sites, CTCs extravasate and colonize in the tissues, where MET is induced.

**Figure 2 cells-12-02260-f002:**
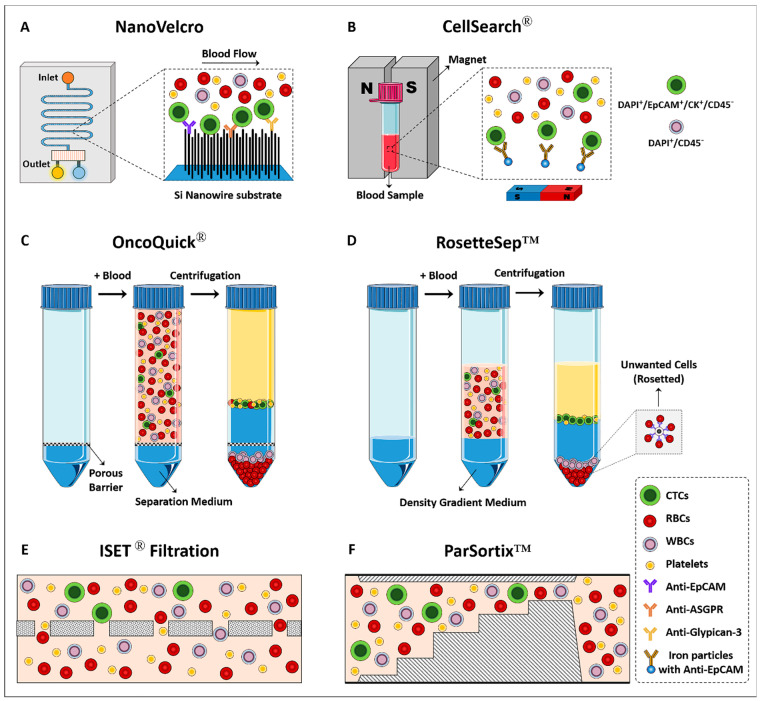
Schematic representation of different methods and technologies that have been developed for the isolation and identification of CTCs.

**Table 1 cells-12-02260-t001:** Summary of single-cell analysis methods used for CTCs.

	Method	Application	Advantages	Disadvantages
Genomic	Pure PCR-based amplification	Amplifying specific sites in the genome	Better uniformity of amplification	Uneven amplification, low coverage, amplification errors, allele dropout
	MDA-based methods in HCC	Point mutations amplifying to analyze the genome of patient-derived CTCs	Higher fidelity than PCR-based methods	Amplification bias, allele dropout
	MALBAC combines MDA and PCR-based methods	Analysis of single-nucleotide variants (SNVs)	Intermediate coverage and uniformity	Allele dropout
	LIANTI	Amplifies T7 promoter-tagged DNA fragments into thousands of RNA copies.	Covers 97% of the genome with a reduced false-negative rate.	
	GenomPlex and Ampli1	Copy-number variation (CNV) profiling	Maintains representation of the entire genome through subsequent reamplifications. Preserves precious source material by amplifying nanogram amounts of starting genomic DNA.	Significantly higher genomic coverage
Transcriptomic	STRT-seq	An established approach to profile entire transcriptomes of individual cells from different cell types	High specificity	5′-only end base
	Smart-seq and Smart-seq2	Single-cell gene expression analyses hold promise for characterizing cellular heterogeneity.	Good coverage of the transcriptome with rarer transcripts being detectable Independent of cell size	High cost, low specificity, low number of cells
	CEL-seq	Single-cell RNA-Seq using multiplexed linear amplification	Sensitive, accurate, and reproducible	3′-only end base, low number of cells
	InDrop and Drop-seq	Sequence thousands of single cells in parallel	Cost benefit, high specificity	3′-only end base
	Mars-seq	Analysis to explore cellular heterogeneity by assembling an automated experimental platform that enables RNA profiling of cells	Long-term storage, cost benefit, high specificity	3′-only end base
	10x Genomics Chromium	A droplet-based scRNA-seq technology allowing genome-wide expression profiling for thousands of cells at once	Cost benefit, high sensitivity and precision	Must process immediately
Epigenomic	sci-ATAC-seq	Generation of sequencing library molecules is selective toward regions of open chromatin on the hyperactive derivative of the cut-and-paste Tn5 transposase	High throughput, independent of antibody	Low coverage per cell
	scChIP-seq	Enabled in-depth characterization of protein-DNA interactions of histone marks at single-cell resolution	High throughput	Low coverage per cell, dependence on antibody

Pure PCR-based amplification (DOP-PCR), multiple displacement amplification [70], degenerate oligonucleotide-primed polymerase chain reaction, multiple annealing- and looping-based amplification cycles (MALBAC), or linear amplification via transposon insertion (LIANTI).

**Table 2 cells-12-02260-t002:** Whole-genome amplification (WGA) methods used for CTC analysis.

Method	Application	Advantage (s)
Pure PCR-based amplification	Amplifying specific sites in the genome	Better uniformity of amplification
MDA-based methods	Point mutations amplification to analyze the genome of patient-derived CTCs	Higher fidelity than PCR-based methods
MALBAC combining MDA and PCR-based methods	Analysis of single-nucleotide variants (SNVs)	Intermediate coverage and uniformity
LIANTI	Amplifies T7-promoter-tagged DNA fragments into thousands of RNA copies	Covers 97% of the genome with a reduced false-negative rate
GenomPlex and Ampli1	Copy-number variation (CNV) profiling	Significantly higher genomic coverage

Pure PCR-based amplification (DOP-PCR), multiple displacement amplification [70], degenerate oligonucleotide-primed polymerase chain reaction, multiple annealing- and looping-based amplification cycles (MALBAC), or linear amplification via transposon insertion (LIANTI).

**Table 4 cells-12-02260-t004:** Clinical applications of some platforms used for the detection of HCC-CTCs.

Platform	Study Group	CTC Positive Detection Rate	Ref.
Cell Search	123 HCC patients; 5 control patients; 10 healthy volunteers	66.67% in patients prior to resection, 28.15% 1 month after resection	[47]
	59 HCC patients; 19 control patients	30.5% in HCC patients	[109]
	20 HCC; 10 control patients	35% in HCC patients	[46]
	21 HCC patients	4.7% in HCC patients	[111]
	57 HCC patients undergoing resection	15% in HCC patients	[112]
	89 HCC patients treated with chemoembolization	56% in HCC patients	[113]
	144 HCC patients	56.9% in patients prior to resection, 30.6% 1 month after resection	[114]
	26 HCC patients	27% in HCC patients	[115]
CanPatrol^TM^	195 HCC patients	95% in HCC patients	[38]
	112 HCC patients; 12 HBV patients; 20 healthy volunteers	90.18% in HCC patients, 16.67% in HBV patients	[93]
	165 HCC patients	70.9% High CTC count was correlated with BCLC stages, multiple tumors, and high levels of alpha-fetoprotein	[116]
	113 HCC patients	78.8%	[117]
	99 HCC patients	89.9%	[118]
	160 HCC patients undergoing resection	90%	[119]
	56 HCC patients	92.86% before liver transplantation surgery	[120]
ISET	7 HCC patients undergoing tumor resection; 8 chronic cirrhosis patients; 8 healthy volunteers	52% in HCC patients	[63]
	44 HCC patients; 30 chronic hepatitis patients; 39 liver cirrhosis patients; 38 healthy volunteers	52% in HCC patients	[64]
RosetteSep	109 HCC patients	92.7% in patients with advanced HCC and candidates for sorafenib treatment	[121]
	32 HCC patients; 17 other types of cancer; 3 acute hepatitis A patients; 6 chronic hepatitis B patients; 4 chronic hepatitis C patients; 15 cirrhosis patients; 12 patients with benign intrahepatic space-occupying lesions	91%	[61]
NanoVelcro	61 HCC patients; 8 healthy control patients	96.7% in HCC patients and 25% in healthy control patients	[57]
CTC^−BioT^Chip	42 HCC patients	59.5%	[53]
OncoQuick	17 HCC patients; 13 healthy volunteers	76.5% in HCC patients	[122]
CytoSorter™	47 HCC patients received PD-1 inhibitor combined with intensity-modulated radiotherapy and anti-angiogenic therapy	95.7%	[123]

BCLC, Barcelona clinic liver cancer; HBV, hepatitis B virus; HCC, hepatocellular carcinoma.

## Data Availability

All data supporting the findings of this study are available within the article or from the corresponding author upon reasonable request.

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
