# Peer review of "Circulating Tumor Cells as a Promising Tool for Early Detection of Hepatocellular Carcinoma"

_cells, 2023, doi:10.3390/cells12182260_

Round 1

Reviewer 1 Report

Authors in this review provide interesting insights into diagnosis of HCC. It will be a plus if authors could bring and cite more updated progress in terms of detection of HCC CTCs by means of varieties of strategies, that will help broaden the vision of readers in the field. One of such novel detection strategies is to phenotypically and karyotypically detect aneuploid CTCs expressing tumor markers by iFISH (Wang et al., 2018 Cancer Lett 412:99).

Author Response

Authors in this review provide interesting insights into diagnosis of HCC. It will be a plus if authors could bring and cite more updated progress in terms of detection of HCC CTCs by means of varieties of strategies, that will help broaden the vision of readers in the field. One of such novel detection strategies is to phenotypically and karyotypically detect aneuploid CTCs expressing tumor markers by iFISH (Wang et al., 2018 Cancer Lett 412:99).

Answer: Thank you very much for your positive feedback and valuable comment. We read the mentioned paper, cited, and discussed in the “Label-dependent strategies” part of the manuscript (pages 4-5) as proposed here below:

“In 2018, a novel subtraction enrichment immunostaining-fluorescence in situ hybridization (SE-iFISH) strategy was developed to detect the HCC-CTCs. This technique was based on comprehensive detection of in situ phenotypic and karyotypic characterisation of hepatocellular CTCs (CD45−/CD31−) in the patients subjected to surgical resection [53, 54]. Besides, microgravity array (MCA) system was also used to detect CTCs and their mRNA expression in HCC patients [55]”.

Reviewer 2 Report

Authors described potential utility of the analysis of CTC for early detection of hepatocellular carcinoma. In my opinion, the paper is too methodological. Authors should rather focus on clinical evidence, and then summarize them in tables and describe to rise a proper conlcusion.

There are minor issues.

Author Response

Authors described potential utility of the analysis of CTC for early detection of hepatocellular carcinoma. In my opinion, the paper is too methodological. Authors should rather focus on clinical evidence, and then summarize them in tables and describe to raise a proper conclusion.

Answer: Thank you for your helpful comment. After a comprehensive literature review, table 4 is inserted now in the revised version of the manuscript (pages 17-18). It lists the clinical studies conducted so far and practically applying some novel CTC-based diagnosis methods.

Reviewer 3 Report

The manuscript summarizes the role of CTCs in clinical decision making strategy in HCC patients. Accordingly, the manuscript is well structured and requires minor comments to be accpeted for the publication

- In the introduction section, please, could underline the critical issues deriving from the analysis of CTC in diagnostic routine cases?

- Table 1, please, could the authors review tabel 1 add a dedicated column with advantage respect applicatio nsection?

- Table 3, please could the authors add a ref for each reporting data?

- Discussion, please, could the authors describe an integrated model where cfDNA and CTCs may  be helpfull in the clinical adminsitration of HCC patients?

Minor editing of English language required

Author Response

In the introduction section, please, could underline the critical issues deriving from the analysis of CTC in diagnostic routine cases?

Answer: Thank you for your helpful suggestion. To address this comment, the following sentences were added to the introduction section (page 2).  

In the clinic, it was shown that CTC-based micro-devices could be an ideal modality for point-of-care testing. However, due to their heterogeneity, reliable detection of CTCs in the body fluids is still a major limitation. Indeed, CTCs derived from different tissues have various characteristics such as different sizes, markers, and immune-phenotyping profiles which make their detection more challenging. Furthermore, several other limiting factors such as damage and fragmentation, both in vivo or in vitro during isolation process, hamper their clinical application.

Table 1, please, could the authors review table 1 add a dedicated column with advantage respect application section?

Answer: Thank you for your comment. One additional column is inserted in the table 1 of the revised version of the ms and entitled “Advantages”.

Table 3, please could the authors add a ref for each reporting data?

Answer: Thank you for your comment. Relevant references have been cited in table 3.

Discussion, please, could the authors describe an integrated model where cfDNA and CTCs may be helpful in the clinical administration of HCC patients?

Answer: Thank you for your comment. The here below proposed paragraph is inserted in the revised version of the ms (page 8):

“Application of cell free DNAs (cfDNAs) as molecular targets can be used for the detection of HCC-CTCs. cfDNA-based technologies have several advantages such as real-time monitoring of the genetic landscape of the tumor, high sensitivity and specificity, and being a non-invasive biopsy procedure. However, the application of this technology has faced some challenges including ensured purity of cfDNA samples, genetic heterogeneity of tumors, and development of standardized protocols for the analysis of cfDNAs data [85].”

And in the conclusion section (page 19)

“Moreover, the combination of CTCs plus cfDNAs detection methods can be a promising approach for early diagnosis.”

Round 2

Reviewer 2 Report

Authors corrected paper according to the addressed comments.